# Cord Blood-Based Approach to Assess Candidate Vaccine Adjuvants Designed for Neonates and Infants

**DOI:** 10.3390/vaccines9020095

**Published:** 2021-01-27

**Authors:** Daisuke Tokuhara, Norikatsu Hikita

**Affiliations:** Department of Pediatrics, Osaka City University Graduate School of Medicine, Osaka 545-8585, Japan; nhikita0104@yahoo.co.jp

**Keywords:** vaccine, adjuvant, children, innate immunity, cord blood, toll-like receptor, zymosan

## Abstract

Neonates and infants are particularly susceptible to infections, for which outcomes tend to be severe. Vaccination is a key strategy for preventing infectious diseases, but the protective immunity achieved through vaccination typically is weaker in infants than in healthy adults. One possible explanation for the poor acquisition of vaccine-induced immunity in infants is that their innate immune response, represented by toll-like receptors, is immature. The current system for developing pediatric vaccines relies on the confirmation of their safety and effectiveness in studies involving the use of mature animals or adult humans. However, creating vaccines for neonates and infants requires an understanding of their uniquely immature innate immunity. Here we review current knowledge regarding the innate immune system of neonates and infants and challenges in developing vaccine adjuvants for those children through analyses of cord blood.

## 1. Introduction

Children, especially neonates and infants, are particularly susceptible to infection and tend to have more severe outcomes than do older children and healthy adults [1,2,3]. Vaccination is the central strategy for providing protective immunity against pathogens [3,4,5,6], and the number of licensed vaccines continues to increase worldwide. For example, only 14 licensed vaccines were available in Japan in 2000, but this number had increased to 26 in 2020. Because most licensed vaccines (e.g., against rotavirus, *Haemophilus influenzae* type b, poliovirus) are used in children, the preventive strategy against pediatric infectious diseases is continually evolving. However, several issues remain to be resolved regarding the use of vaccines in children. First, because of their immature immune systems, children—particularly infants—often require multiple vaccine doses to achieve protective immunity [3,6,7]. Increasing the number of necessary vaccinations associated with multiple doses will be burdensome for children and their families. In addition, internationally licensed vaccines are as yet unavailable for several common respiratory and gastrointestinal pathogens, such as respiratory syncytial virus (RSV) and norovirus, which cause high mortality and hospitalization rates in infants and sometimes lead to high morbidity, especially in developing countries. The previously licensed RSV vaccine (a formalin-inactivated whole-virus vaccine) developed in the 1960s induced several fatal cases of vaccine-enhanced disease (e.g., eosinophilic pneumonia) in infants and therefore has been withdrawn from the market [8,9]. For rotavirus, there are currently two licensed vaccines (Rotateq and Rotarix) with safety and efficacy profiles in infants [10,11], however a previously licensed product (RotaShield) developed in the 1990s induced intussusception events in vaccinated infants, leading to its withdrawal [12]. Therefore, there is great demand for pediatric vaccines that are both effective and safe in neonates and infants.

During the traditional process of vaccine development, effectiveness typically is assessed first by using adult animals (e.g., adult mice, and pigs), which therefore have mature immune systems [13,14,15,16], followed by further evaluation in healthy adult humans and sometimes infant animals (e.g., infant rhesus monkeys) [17,18,19]. Finally, candidate vaccines are assessed through clinical trials in children [20,21]. Consequently, current vaccine development primarily is based on results from adult humans and animals and does not reflect the features of the immune system of human children [22,23,24]. Therefore, the development protocol for vaccines to be used in neonates and infants needs to be revised to incorporate an understanding of their immune system.

## 2. Infant Immune Development and Its Impact on Vaccine Response

In terms of the gastrointestinal and respiratory tracts—the infection sites for most of the important pathogens for children—the gut-associated lymphoid tissue (GALT) and nasopharyngeal-associated lymphoid tissue (NALT) play central roles in the induction of protective immunity (e.g., mucosal secretory IgA and systemic serum IgG) against pathogens and vaccine antigens at the mucosa [3,25,26]. Those mucosal immune systems develop both anatomically and functionally in an age-dependent manner in utero and during childhood [27,28,29] (Figure 1). In particular, the Peyer’s patches (PPs) and mesenteric lymph nodes (MLNs) of GALT begin developing in utero, whereas isolated lymphoid follicles of the small intestine and NALT develop and mature after birth [25,30,31]. In humans, aggregates of T lymphocytes that contain small numbers of B cells are the initial structures of PPs and are present beginning at 14 weeks of gestational age [32]. PPs increase in number and size with age until adolescence [33].

In murine models, the underdevelopment of MLNs, PPs, or both coincides with a reduction in secretory IgA production and thus an increase in susceptibility to infection [34]. In a previous study, MLN- and PP-deficient mice showed strong reductions in IgA- producing B-cell counts and increased susceptibility to *Listeria monocytogenes* infection [34]. In another study, oral immunization with tetanus toxoid and cholera toxin as mucosal adjuvant failed to induce tetanus toxoid-specific intestinal IgA responses in PP- and MLN-deficient mice [35]. These findings suggest that the presence and maturation of mucosa-associated lymphoid tissues play key roles in the induction of effective, antigen-specific, protective immunity due to vaccination.

In addition to the anatomical development of immune inductive tissues, the functional maturation and development of immune cells (dendritic cells [DCs], monocytes, B and T cells) and maternal antibodies are important factors to consider regarding vaccine effectiveness in children [36,37,38,39,40,41] (Figure 1). Several studies demonstrated the intrinsic insufficient or immature function of immune cells regarding the induction of acquired immunity by two years of age [36,37,38,39,40]. B-cell activation due to bacterial capsular polysaccharides requires co-stimulation by CD21. However, because neonatal B cells have decreased CD21 expression, newborns and infants are less capable than adults of producing specific antibodies against bacterial capsular polysaccharides [36]. In terms of T cells, infant T cells became primed by measles antigen despite the presence of passive antibodies, whereas their adaptive immune responses (e.g., interferon [IFN]-γ) were poor compared with those of adults [37]. Another study demonstrated the immaturity of neonatal T-cell function by showing that neonatal immunization with oral polio vaccine induced the production of high titers of neutralizing antibodies but reduced proliferative and IFN-γ responses to polio antigens compared with those of adults [38]. Furthermore, T-cell memory is immature in infants immunized against tetanus and diphtheria [39]. Antigen-presenting cells (APCs; e.g., DCs and monocytes)—critical immune cells that bridge between innate immunity and acquired immunity—develop during the first year of life. A previous study showed that the surface expression of CD80 and HLA-DR reaches adult levels within the first three months of life for myeloid DCs and during the first 6–9 months for monocytes and plasmacytoid DCs [40]. Another study described that the expression of CD86, a co-stimulatory molecule, is at low levels on conventional DCs in the draining mediastinal lymph nodes of neonatal compared with adult mice, thus resulting in insufficient induction of RSV-specific CD8+ T cells [42]. Passive antibodies (maternal antibodies) protect offspring from infections but are considered to limit their vaccine responses. Underlying mechanisms of maternal antibodies-mediated inhibition of vaccine responses in the offspring are not fully understood, however a recent study demonstrated, by using influenza hemagglutinin (HA) as vaccine antigen, that maternal antibodies do not prevent B cell activation or germinal center formation, but control plasma cell and memory B cell differentiation, shaping the long-term antigen-specific B cell repertoire [41]. In terms of T cell, maternal antibodies do not affect total CD4+ T cell responses but limit the expansion of T follicular helper cells [41]. On the other hand, another recent study demonstrated by using *Bordeterlla pertussis* as a vaccine antigen that maternal antibodies do not affect the activation and proliferation of CD4+ and CD8+ T cells in their neonatal cord blood after *Bordeterlla pertussis* stimulation [43].

Not only intrinsic immaturity but also nutrient deficiency (e.g., vitamins A and D) diminishes immune cell function in regard to the vaccine-mediated induction of antibody, especially the mucosal IgA response [44,45,46,47]. Vitamin A deficiency impaired the specific intestinal IgA response to RotaTeq, the oral pentavalent rotavirus vaccine, in neonatal gnotobiotic pigs [44]. Another study demonstrated impairment of the specific intestinal IgA response to oral cholera vaccine in vitamin A-deficient mice [45]. In the respiratory immune system, vaccine antigen-specific IgA antibody-forming cells in cervical lymphoid nodes or respiratory tract-associated tissues were specifically decreased after intranasal immunization of vitamin A-deficient mice [46]. In the induction of the gut mucosal IgA response, intestinal DC-derived retinol, a metabolite of vitamin A, enhances the expression of α4β7 integrin and CC chemokine receptor 9 (CCR9) on T and B cells upon activation and imprints them with ‘gut-homing’ specificity to the intestinal lamina propria [48]. Due to gut-homing, the activated T and B cells migrate from PPs through the efferent lymphatics to the regional mesenteric lymph nodes and then to the intestinal lamina propria [28,49]. Therefore, vitamin A deficiency significantly decreases subsequent gut homing of T and B cells due to lack of the antigen-specific intestinal IgA response. In clinical settings, both the Rotateq and Rotarix vaccines have been shown to be safe, immunogenic, and highly protective against severe rotavirus gastroenteritis in infants in developed countries (e.g., North America and Europe); however, their effectiveness was much lower in Africa and Asia [50,51]. Because vitamin A deficiency impairs the induction of acquired immune responses to those live-attenuated rotavirus vaccines in animal model [52], nutritional insufficiency (e.g., vitamin A) may be one of the hypothesized mechanisms on the reduced vaccine efficacy in infants in developed countries. On the other hand, a recent study demonstrated a genotype-dependent manner in the susceptibility to rotavirus infection and reported that the Lewis A phenotype is a restriction factor for Rotateq and Rotarix vaccine responses in Nicaraguan children [53]. In addition, another study elucidated that environmental enteropathy and malnutrition were associated with oral vaccine (Rotarix and Polio) failure in infants in Bangladesh [54]. Taken together, various factors (e.g., vitamin A deficiency, genotype, and the burden of environmental enteropathy) need to be taken into consideration for the use of vaccine in infants in developed countries.

Due to this anatomic and functional immaturity, which sometimes is exacerbated by nutritional deficiency, the immune systems of neonates and infants tend to respond poorly to pathogens and current vaccines [37,55,56] (Figure 1). The development of vaccine adjuvants that are effective in children will require novel approaches that specifically address immature immune systems and that test candidates in neonatal and infant animals and humans. In addition, although the use of pathogen subunits ( e.g., cholera toxin B subunit and virus-like particles), peptides, mRNA, or DNA as vaccine antigens has the potential to accelerate vaccine development [3,14,57,58], the resulting vaccines ( e.g., those using recombinant protein) present a potential risk of reduced immunogenicity; this situation thus could be compounded in neonates and infants, who are expected to respond even less robustly to vaccine antigens (e.g., recombinant protein) compared with the whole pathogen. Therefore, new technology that enhances the immune responses against vaccine antigens in neonates and infants is needed. In this regard, because vaccine adjuvants enhance antigen recognition by host APCs, such as DCs, and thus contribute to the effective induction of protective acquired immunity [59,60,61], the development of vaccines for children must include appropriate adjuvants that efficiently activate the immature immune system. To date, various adjuvants (e.g., alum, chitosan, and MF59) have been developed and used in mice and humans [62,63,64,65,66,67], and some of them were tested in children [64,66,67]. Alum, salts of aluminum, is the classical adjuvant most often used in vaccines in humans. A recent study used alum as an adjuvant for enterovirus 71 vaccine in healthy children aged 6–35 months and reported the safety and effectiveness of the alum-adjuvant vaccine [64]. On the other hand, another study demonstrated in vitro that an adjuvant effect of alum was weak in neonatal DCs compared with those of BCG and TLR8 agonist [68]. MF59, a potent oil-in-water emulsion, has been developed and used as adjuvant for influenza vaccine and its safety and effectiveness confirmed in adult [65]. Recent studies also demonstrated that MF59 is effective and safe in young children 6–35 months as influenza vaccine adjuvant [66,67], whereas it needs further confirmation whether MF59 is safe and effective in children as a potent universal adjuvant for other vaccine antigens.

## 3. Human Cord Blood-Based Approach to Assess Candidate Vaccine Adjuvants Designed for Neonates and Infants

Developing vaccine adjuvants for use in children, especially neonates and infants, involves understanding the unique aspects of their immature immune systems. In this regard, immune cells (e.g., DCs and monocytes) and tissues (e.g., PPs and MLNs) obtained from human children are most advantageous for immunologic research. However, collecting sufficient volumes of blood or other tissues from neonates and infants is excessively invasive compared with adults and thus typically is unacceptable from an ethical standpoint. To overcome these hurdles, previous studies took an alternative approach by collecting human cord blood as a substitute for peripheral blood from neonates and infants [40,69,70,71,72,73,74] (Figure 2). Because it is obtained from the delivered placenta (ex utero), which is usually discarded after childbirth, human cord blood can be collected without causing pain or anemia in mothers and their babies and therefore can be considered as a cost-effective, humane resource for investigations of human neonatal and early infantile immunity. Approximately 30 to 50 mL of cord blood can be harvested from a human placenta and used as a rich source of APCs (e.g., monocytes and DCs). For example, 1 to 2 × 10^8^ monocytes can be isolated from human cord blood (50 mL) through positive selection using CD14 magnetic microbeads [69]. These isolated and/or in vitro differentiated APCs (e.g., monocytes, DCs, and monocyte-derived DCs [MoDCs]) can be applied to in vitro stimulation experiments under various conditions [69,70,71] (Figure 2).

Compared with adults, healthy newborns have less experience with infections and lack pre-existing immunologic memory; therefore protective immunity in neonates largely relies on the innate immune system and IgG antibodies that have been transferred from their mothers via the placenta [75,76,77]. In addition, the T and B cells that confer pathogen- and antigen-specific protection generally require activation by APCs (e.g., DCs), which themselves are stimulated by pathogens or various antigens via the innate immune system [78]. Consequently, the immune cells, especially APCs (e.g., monocytes and DCs), in human cord blood after a healthy delivery (i.e., in the absence of infection, hypoxia and severe maternal complications) are ideal materials for studying the innate immune system of human neonates and infants.

In addition, by recognizing pathogen-associated molecular patterns on APCs, Toll- like receptors (TLRs) play a key role in innate immunity, and previous studies exploring immature innate immunity focused on comparing TLR signaling between human cord blood-derived immune cells and adult peripheral blood-derived immune cells [75,78]. One advantage of using human cord blood cells is the opportunity to obtain immunologic data pertinent to the human innate immune system. Previous studies showed that TLR function and regulation differ between human and other animal immune cells [79,80,81,82,83]. For example, *Rhodobacter sphaeroides* acts as an agonist of TLR4 signaling in horses and hamsters but as an antagonist in humans and mice [81]. TLR5 shows species-specificity toward different bacterial flagellins; for example, the flagellin of *S. enterica* serovar Typhimurium activated chicken TLR5 more strongly than human TLR5 but not mouse TLR5 [82]. Regarding TLR3, a receptor for viral dsRNA, previous findings suggest differences in viral susceptibility between TLR3-deficient mice and humans [83]. Therefore, the immunological findings obtained by using human cord blood-derived immune cells (e.g., DCs and monocytes) better reflect the features and characteristics of the human immune system compared with those of non-human animals and thus are more relevant to human clinical trials. Previous human cord blood-based studies consistently demonstrated that TLR-mediated neonatal innate immunity is selectively impaired compared with that of adult immune cells [40,69,70,72,73]; however, the details of this selective impairment of TLR-mediated immune response vary between studies [40,69,70,72,73]. For example, one study revealed that the responses of monocytes to TLR1/2, TLR2/6, TLR4, or TLR7 stimulation were lower in human cord blood than in adult blood, whereas TLR7/8 stimulation elicited comparable immune responses in human cord blood- and adult peripheral blood- derived monocytes [72]. Another study revealed that cytokine production in response to TL4 stimulation (TNF-α, CXCL10, IL−12 p70, and IFN-γ) and TLR9 activation (CXCL10 and CXCL9) was significantly reduced in human cord blood cells compared with adult peripheral blood cells [40]. Preterm delivery further influences impairment of neonatal innate immunity. TLR9-mediated IFN-α production in plasmacytoid DCs was markedly lower in preterm newborns than in term newborns and adults [84].

These selective impairments of the TLR-mediated innate immune response prompted us to use human cord blood-based analyses to identify a TLR agonist that effectively stimulated the neonatal innate immune system; such a reagent might be an ideal vaccine adjuvant for neonates and infants [69,70] (Figure 2). In this regard, we first found that the immature TLR-mediated innate immunity in neonates depends on monocytes rather than DCs [69]. The production of inflammatory cytokines (e.g., interleukin [IL]−6 and IL−8) in response to stimulation with lipopolysaccharide (TLR4 agonist), Pam3 CSK4 (TLR1/2 agonist), flagellin (TLR5 agonist), and polyinosinic:polycytidylic acid (poly[I:C]; TLR3 agonist) was generally weaker in monocytes, comparable in DCs, and greater in MoDCs derived from human cord blood than adult peripheral blood counterparts [69]. In contrast, we found that zymosan, a cell-wall extract from *Saccharomyces cerevisiae* that is composed mainly of β-glucan, induced TLR2/6 heterodimer-mediated inflammatory cytokine production (IL−6, IL−8, and TNF-α) that was comparable among the monocytes, DCs, and MoDCs of both cord and adult blood [69]. In the induction of vaccine-mediated protective immunity, DCs are important APCs, whereas the role of MoDCs in immunity induction is not fully understood. Because MoDCs are inflammatory DCs and involved in the pathogenesis of autoimmune diseases, the immune response of MoDCs to immunization may lead to unfavorable side effects in neonates.

We therefore consider zymosan a promising vaccine adjuvant for use in children because zymosan can be expected to achieve vaccine-mediated immune responses in neonates and infants comparable to those in adults. In other words, information regarding the safety, adverse events, and effectiveness of vaccines using zymosan as an adjuvant obtained from adult volunteers likely will be similar for infants. Therefore unexpected adverse events regarding the use of these vaccines in infants can be avoided. In terms of the usefulness of β-glucan, another group’s experiments demonstrated that *Candida albicans*-derived β-glucan, similar to that in zymosan, induced immune responses (e.g., IL−6, IL−10, and TNF-α levels) from cord blood-derived monocytes that were similar to those of adult peripheral blood-derived monocytes [85]. Together these results suggest that β- glucan (whether derived from *S. cerevisiae* or *C. albicans*) could be an appropriate candidate vaccine adjuvant for neonates and infants.

In the induction of acquired protective immunity, antigen–MHC II complexes and CD80 and CD86 costimulatory molecules on APCs (e.g., DCs) activate and expand helper T cells, which then induce B cell growth and antigen-specific antibody production. Regarding these antigen-presentation molecules, we elucidated significantly lower basal levels of expression of HLA class II and CD80 in cord blood monocytes than adult monocytes, thus confirming selectively impaired antigen-presentation ability in neonatal APCs [70]. In contrast, zymosan enhanced the expression of HLA class II and CD86 to greater levels in cord blood monocytes than in adult peripheral blood monocytes [70]. These results support the effectiveness of zymosan as a vaccine adjuvant for children via the enhancement of vaccine antigen presentation by APCs.

Another important aspect to consider regarding candidate vaccine adjuvants for use in neonates and early infants is the influence of maternal factors. Maternal complications (e.g., diabetes mellitus, vitamin D deficiency, alcohol intake) are well known to alter the innate immune system and susceptibility to infection [71,86,87,88,89,90]. Elevated levels of T- helper type 1 cytokines and low levels of IL−10 have been reported in the serum of macrosomic babies born to mothers with gestational diabetes mellitus [87]. Other studies have demonstrated that maternal alcohol intake and vitamin D deficiency significantly exacerbate neonatal and infantile infections, thus suggesting impairment of neonatal immunity [88,89], and that maternal smoking impairs the TLR-mediated innate immune response in neonates [90]. To obtain the stable immunostimulatory effects of zymosan as a candidate vaccine adjuvant for neonates and infants, it is important to know the influence of maternal factors on zymosan-mediated neonatal innate immunity. In this regard, we have confirmed that maternal diabetes mellitus, the most frequent pregnancy-related complication, exacerbated the elevation of IL−8, TNF-α, or both through stimulation by Pam3 CSK4 (TLR1/2 heterodimer agonist) or flagellin (TLR5 agonist) but has no effect on zymosan-mediated innate immune responses in neonates [71]. Those results further strengthen the notion that zymosan could be an effective, safe, and stable vaccine adjuvant for neonates and infants.

The potential usefulness of zymosan, an agonist of the TLR2/6 heterodimer, as a systemic or mucosal (intranasal or oral) vaccine adjuvant has been evaluated in several adult animal studies (mouse, chicken and pig) with mature immune system [91,92,93,94]. A few studies also demonstrated the potential usefulness of other TLR2/6 heterodimer agonists Pam2 CSK4 and dipalmitoyl lipopeptide as vaccine adjuvants in adult animal mice in vitro or in vivo [95,96]. Although these studies using adult animals suggest the potential usefulness of TLR2/6 heterodimer agonists as vaccine adjuvants, whether zymosan, Pam2 CSK4 and dipalmitoyl lipopeptide likewise induce effective immune responses in neonatal and infantile immature immune systems is still unclear. To address this issue, we compared the immune responses to 2 TLR2/6 heterodimer agonists (zymosan and macrophage-activating lipopeptide 2 [MALP2]) in cord blood-based analyses [97]. We found that MALP2 induced cytokine production (e.g., IL−8 and IL−6) did not differ between monocytes and DCs derived from cord blood compared with adult peripheral blood, whereas MALP2-mediated cytokine production was elevated in cord blood-derived MoDCs [97].

Both zymosan and MALP2 are considered to activate innate immunity via the TLR2/6 heterodimer, but their co-stimulatory molecules differ. Zymosan acts through CD14 and dectin−1 to achieve TLR2/6-mediated stimulation [98,99], whereas MALP2 uses CD36 and RP105 to this end [100,101,102]. These differences in required co-stimulatory molecules between zymosan and MALP2 may underlie the different features of the resulting immune responses. Because MoDCs can be involved in both inflammatory and autoimmune reactions [103,104,105,106], the excessive immune response triggered by MALP2 in cord blood-derived MoDCs compared with those from adults is undesirable in vaccine adjuvant development for neonates and infants. An in vitro human cord blood study demonstrated that the stimulation of neonatal MoDCs via dectin−1 and TLRs induced IL−12 p70 secretion and the Th1 polarization of neonatal T cells, thus implicating dectin−1 agonist as a Th1 adjuvant [107]. Another study indicated that zymosan-mediated DNA vaccination enhanced Th1-mediated immunity [91]. Because RSV vaccine-enhanced disease (e.g., eosinophilic pneumonia) is considered to be a Th2-driven pathology [108], zymosan—as a TLR2/6 and dectin−1 agonist—might be used to develop a safe RSV vaccine via the regulation of Th2-associated vaccine-enhanced disease. Potential usefulness of zymosan as an adjuvant for neonates and infants needs to be further evaluated in the future studies.

## 4. Future Perspectives

Because it is an in vitro technique, one limitation of the human cord blood-based approach to developing vaccine adjuvants is the inability to directly elucidate the clinical effectiveness (e.g., specific antibody production and protection against pathogens) of those vaccine adjuvant candidates. Future immunization studies in animal models (e.g., neonatal mice) or human adults will be needed to confirm the safety and effectiveness of vaccine antigens (e.g., norovirus virus-like particles and RSV G or F protein) administered with candidate vaccine adjuvants (e.g., zymosan) identified through cord blood-based investigations. Given their safety and effectiveness, these vaccine–adjuvant combinations are expected to proceed to clinical trials in infants. In addition, because RSV leads to the acute lower respiratory tract infection in early infancy, the induction of protective immunity against RSV via neonatal immunization is an important strategy. Therefore, the development of an RSV vaccine specifically for neonates is a future topic of interest.

Major similarities and differences among neonatal and infant animal models well used for the immunization research are summarized in Table 1. Based on the understanding of those similarities and differences, it is necessary to plan the immunization protocols. For example, neonatal and infant period is exceedingly short in mice, rats, and piglets compared with human. Weaning period is 3–4 weeks in rodents and piglet but is six months in human [109,110,111]. It is general for human infants to plan the repeated immunization protocol with a 3–4 week time interval, but it is difficult to do the same immunization protocols during the relatively short neonatal and infant period in rodents and piglets. In the case in which researchers use the repeated immunization schedule for neonatal mice, the neonatal murine immunization study may finish it’s boost immunization in the adult period thus it may not accurately reflect neonatal or infant immunization. In contrast, non-human primates provide a clinically relevant model that more closely resembles human immunologically and genetically compared with other well-used animal models (e.g., rodents and piglets) [112]. The weaning period of non-human primates (e.g., macaque) is 5–10 months [113], that is significantly longer than rodents and piglets, thus immunization protocols similar to human are available [114]. The use of the neonatal and infant animal models is an undoubtfully a useful approach for neonatal and infant vaccine development, but it is important to carefully interpret the results of neonatal and infant animal immunizations according to the characteristics of the used animal models when these results are applied to human clinical trials.

## 5. Conclusions

No licensed vaccine currently available was developed according to the particular features of the neonatal and infant immune system. Because the innate immune systems of neonates and infants are developmentally incomplete and thus immature compared with those of healthy adults, current vaccine development strategies based on the mature immune system (e.g., of adult humans) need to shift to address the unique challenges presented by this immunologic immaturity. In this regard, a human cord blood-based approach for screening candidate vaccine adjuvants is non-invasive and cost-effective and reflects the neonatal innate immune system; such a system may become key to developing safe and effective vaccines for children.

## Figures and Tables

**Figure 1 vaccines-09-00095-f001:**
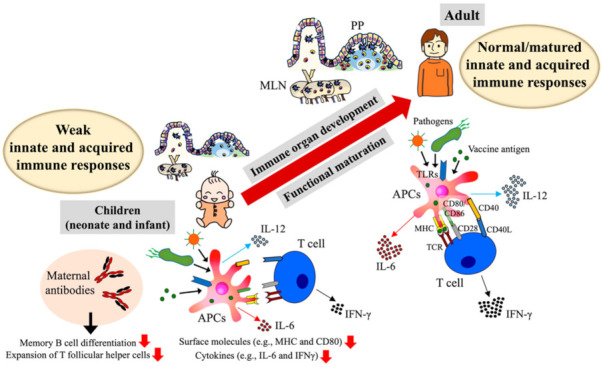
Infant immune development and its impact on vaccine response. Immune inductive tissues (e.g., Peyer’s patches [PP] and mesenteric lymph node [MLN]) and immune cells (e.g., antigen-presenting cells [APCs] such as dendritic cells) age-dependently develop after birth to adulthood. Immaturity of immune inductive tissues and immune cells, and the presence of maternal antibodies are responsible for the weak innate and acquired immune responses in neonates and infancy. Upon encountering pathogens or vaccine antigens, toll-like receptors (TLRs)-mediated immune responses are weak in neonatal and infant immune cells. Among immune cells, neonatal and infant APCs are less capable of expressing surface molecules (e.g., MHC class II antigen and CD80) related to the antigen presentation and producing cytokines (e.g., interleulin [IL−12] and IL−6) related to antigen-specific B and T cells maturation compared with those of adult. Neonatal T cells show reduced production of IFNγ upon vaccination.

**Figure 2 vaccines-09-00095-f002:**
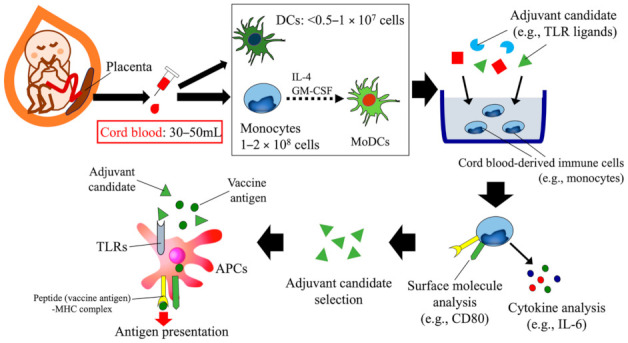
Concept of the human cord blood-based development of vaccine adjuvant candidate. Immune cells (monocytes, DCs, and MoDCs) derived from cord blood can be used to study the innate system of human neonates. These cells can be used in in vitro tests to assess the ability of various substances to induce favorable immune responses (e.g., IL−6 production and CD80 expression). In particular, these tests can identify neonate- and infant-specific adjuvant candidates that effectively activate APCs to induce vaccine antigen-specific protective immunity. DC, dendritic cell; MoDC, monocyte-derived dendritic cell; TLR, toll-like receptor; APC, antigen-presenting cell; IL−6, interleukin−6.

**Table 1 vaccines-09-00095-t001:** Major similarities and differences among neonatal and infant animal models for the immunization research.

Major similarities	Overall reduced expressions of surface molecules (e.g., MHC class antigens, CD80, CD86 and/or CD40) and low induction of cytokines (e.g., IL−6 and IL−12) on APCs [40,42,69,70,115]. Reduced induction of IFNγ on T cells [37,116]
	Immaturity of immune organs (e.g., PPs and MLNs) [26,27,28,115,117]
Major differences	Species-specific TLR expressions [118,119,120,121] (e.g., human, TLRs 1–10; mouse and rat, TLRs 1–9 and 11–13; piglet, TLRs 1–10)
	Species-specific TLR responses [122,123] (e.g., LPS sensitivity, human > piglet > mouse and rat)
	Maternal antibodies [115,124,125] (e.g., human, placentally transferred serum IgG Ab and breast milk IgA Ab; mouse and rat, breast milk IgG and IgA Abs, and placentally transferred serum IgG Ab; piglet, breast milk IgG and IgA Abs)
	Weaning period [109,110,111] (e.g., human, 6 months; mouse, rat and piglet, 3–4 weeks)

## Data Availability

Not applicable.

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
