# Peer review of "Cord Blood-Based Approach to Assess Candidate Vaccine Adjuvants Designed for Neonates and Infants"

_vaccines, 2021, doi:10.3390/vaccines9020095_

Round 1

Reviewer 1 Report

  1. “another study, oral immunization with tetanus toxoid with cholera toxin as mu-70 cosal adjuvant failed to induce tetanus toxoid-specific intestinal IgA responses in PP- and 71 MLN-deficient mice treated in utero with antibodies to TNF receptor 55 and lymphotoxin- β receptor” Please revise this sentence.
  2. “terms of T cells, infant T cells became primed by measles anti-84 gen despite the presence of passive antibodies” How passive antibodies affect T cell priming ?
  3. “Antigen-presenting cells ( APCs; e.g., DCs and mon-ocytes )—critical immune cells that bridge between innate immunity and acquired im munity—develop during the first year of life.” How the immunization in first few months work should be explained here. Also include the references.
  4. Authors should consider adding a figures discussing various immune cells and important markers related to neonatal immune response to infection and vaccines.
  5. Specific sub section title names can be improved. Eg. “Development of Immune system”
  6. “ respiratory tract associated lymphoid tissues ( e.g., nasal wash, NALT, and cervical lymph nodes” Cervival lymph nodes or respiratory tract associated tissues ?
  7. much lower in Africa and Asia [45, 46], thus perhaps explaining 119 the negative effect of nutritional insufficiency (e.g., vitamin A) on vaccine efficacy in in-120 fants in developed countries.” There are many vitamins deficient in these populations. These populations are genetically distinct from North American individuals. Also the disease burden and exposure could be different. Please discuss.
  8. “Potential Usefulness of Zymosan as a Vaccine Adjuvant” This section is not related to neonatal immunity therefore should not be part of this manuscript.
  9. In the conclusion authors discuss use of mice models. Authors should provide a table indicating major similarities and differences in immunity of neoanatal animal models.
  10. There are only 4 papers that cite recent literature. Please check the recent literature and focus on covering recent literature too.

Author Response

Reviewer 1

<Comment 1>  “another study, oral immunization with tetanus toxoid with cholera toxin as mucosal adjuvant failed to induce tetanus toxoid-specific intestinal IgA responses in PP- and MLN-deficient mice treated in utero with antibodies to TNF receptor 55 and lymphotoxin- β receptor” Please revise this sentence.

<Reply to the comment> According to the comment, we revised the sentences (line 72).

<Comment 2>  “terms of T cells, infant T cells became primed by measles antigen despite the presence of passive antibodies” How passive antibodies affect T cell priming ?

<Reply to the comment> The detailed underlying mechanisms of maternal antibodies-mediated inhibition of vaccine responses during infancy is not fully understood, thus we put the word “despite the presence of passive antibodies”. In order to further explain the current knowledge about the relationship between maternal antibodies and vaccine responses, we newly added the sentences as following (line 100-166):

“Passive antibodies (maternal antibodies) are important factor to be taken into consideration for vaccine responses during neonatal and infant period. Maternal antibodies protect offspring from infections but are considered to limit their vaccine responses. Underlying mechanisms of maternal antibodies-mediated inhibition of vaccine responses in the offspring are not fully understood, however a recent study demonstrated by using influenza hemagglutinin (HA) as vaccine antigen that maternal antibodies do not prevent B cell activation or germinal center formation, but control plasma cell and memory B cell differentiation, shaping the long-term antigen-specific B cell repertoire [41]. In terms of T cell, maternal antibodies do not affect total CD4+ T cell responses but limit the expansion of T follicular helper cells [41]. On the other hand, another recent study demonstrated by using Bordeterlla pertussis as a vaccine antigen that maternal antibodies do not affect the activation and proliferation of CD4+ and CD8+ T cells in their neonatal cord blood after Bordeterlla pertussisstimulation [43].”

  1. Vono M.; Eberhardt C.S.; Auderset F.; Mastelic-Gavillet B.; Lemeille S.; Christensen D.; Andersen P.; Lambert P.H.; Siegrist C.A. Maternal Antibodies Inhibit Neonatal and Infant Responses to Vaccination by Shaping the Early-Life B Cell Repertoire within Germinal Centers. Cell Rep 2019, 28, 1773-1784. doi: 10.1016/j.celrep.2019.07.047.

  1. Lima L.; Molina M.D.G.F.; Pereira B.S.; Nadaf M.L.A.; Nadaf M.I.V.; Takano O.A.; Carneiro-Sampaio M.; Palmeira P. Acquisition of specific antibodies and their influence on cell-mediated immune response in neonatal cord blood after maternal pertussis vaccination during pregnancy. Vaccine 2019, 37, 2569-2579. doi: 10.1016/j.vaccine.2019.03.070.

<Comment 3>  “Antigen-presenting cells ( APCs; e.g., DCs and mon-ocytes )—critical immune cells that bridge between innate immunity and acquired im munity—develop during the first year of life.” How the immunization in first few months work should be explained here. Also include the references.

<Reply to the comment> In the first few months, vaccine responses are mainly influenced by the intrinsic immaturity of immune cells and the presence of maternal antibodies. We added the influence of the maternal antibodies on the vaccine responses as described in the reply to the comment 2 (line 79 and lines 100-166).

<Comment 4>  Authors should consider adding a figure discussing various immune cells and important markers related to neonatal immune response to infection and vaccines.

<Reply to the comment> According to the comment, we added the Figure 1 in the revised manuscript.

<Comment 5>  Specific sub section title names can be improved. Eg. “Development of Immune system”

<Reply to the comment> We appreciate the reviewer's comment. We changed the title as following: "Infant Immune Development and its impact on vaccine response" (line 55).

<Comment 6>  “ respiratory tract associated lymphoid tissues ( e.g., nasal wash, NALT, and cervical lymph nodes” Cervical lymph nodes or respiratory tract associated tissues ?

<Reply to the comment> According to the comment, we modified the sentences (line 183).

<Comment 7>  much lower in Africa and Asia [45, 46], thus perhaps explaining the negative effect of nutritional insufficiency (e.g., vitamin A) on vaccine efficacy in infants in developed countries.” There are many vitamins deficient in these populations. These populations are genetically distinct from North American individuals. Also the disease burden and exposure could be different. Please discuss.

<Reply to the comment> We deeply appreciate the reviewer's constructive comments. We modified and newly added the possible influence of genetic and underlying disease burdens on the reduced effects of rotavirus vaccine by citing several literatures as following (lines 203-213):

In clinical settings, both the Rotateq and Rotarix vaccines have been shown to be safe, immunogenic, and highly protective against severe rotavirus gastroenteritis in infants in developed countries (e.g., North Americaand Europe); however, their effectiveness was much lower in Africa and Asia [50, 51]. Because vitamin A deficiency impairs the induction of acquired immune responses to those live-attenuated rotavirus vaccines in animal model [52], nutritional insufficiency (e.g., vitamin A) may be one of the hypothesized mechanisms on the reduced vaccine efficacy in infants in developed countries. On the other hand, a recent study demonstrated a genotype-dependent manner in the susceptibility to rotavirus infection and has reported the Lewis A phenotype is a restriction factor for Rotateq and Rotarix vaccine responses in Nicaraguan children [53]. In addition, another study elucidated that environmental enteropathy and malnutrition were associated with oral vaccine (Rotarix and Polio) failure in infants in Bangladesh [54]. Taken together, various factors (e.g., vitamin A deficiency, genotype and the burden of environmental enteropathy) need to be taken into consideration for the use of vaccine in infants in developed countries.

<Added new references>

  1. Chattha K.S.; Kandasamy S.; Vlasova A.N.; Saif L.J. Vitamin A deficiency impairs adaptive B and T cell responses to a prototype monovalent attenuated human rotavirus vaccine and virulent human rotavirus challenge in a gnotobiotic piglet model. PLoS One 2013, 8, e82966. doi: 10.1371/journal.pone.0082966.

  1. Bucardo F.; Nordgren J.; Reyes Y.; Gonzalez F.; Sharma S.; Svensson L. The Lewis A phenotype is a restriction factor for Rotateq and Rotarix vaccine-take in Nicaraguan children. Sci Rep 2018, 8, 1502. doi: 10.1038/s41598-018-19718-y.

  1. Naylor C.; Lu M.; Haque R.; Mondal D.; Buonomo E.; Nayak U.; Mychaleckyj J.C.; Kirkpatrick B.; Colgate R.; Carmolli M.; Dickson D.; van der Klis F.; Weldon W.; Steven Oberste M.; PROVIDE study teams.; Ma J.Z.;Petri W.A. Jr. Environmental Enteropathy, Oral Vaccine Failure and Growth Faltering in Infants in Bangladesh. EBioMedicine 2015, 2, 1759-66. doi: 10.1016/j.ebiom.2015.09.036.

<Comment 8>  “Potential Usefulness of Zymosan as a Vaccine Adjuvant” This section is not related to neonatal immunity therefore should not be part of this manuscript.

<Reply to the comment> As the reviewer suggested, "Potential usefulness of Zymosan as a vaccine adjuvant" is not directly related to the neonatal immunity, on the other hand, the discussion about Zymosan is our important issue for our own past and ongoing neonatal and infant vaccine adjuvant research. So we have shorten the sentences and inserted the section of " Human cord blood-based approach to assess candidate vaccine adjuvants designed for neonates and infants " (lines 466-585).

<Comment 9>  In the conclusion authors discuss use of mice models. Authors should provide a table indicating major similarities and differences in immunity of neonatal animal models.

<Reply to the comment> We deeply appreciate the reviewer’s constructive comments. We added the simple table for the understanding of the similarities and differences in neonatal and infant animal models for the immunization research at the end of the subsection “Future perspectives”. We also added references for the Table.

<Comment 10>  There are only 4 papers that cite recent literature. Please check the recent literature and focus on covering recent literature too.

<Reply to the comment> According to the comment, we added 10 recent references published since 2015 and added sentences by citing those references (Ref No, 41,43,53,54,62,66,67,68,115,120).

Reviewer 2 Report

Very well written and thoroughly review. The most important point is the author should describe more on commercially available adjuvants, have these been done?

1. Line 48-49 does not reflect the features of the immune system of human children. Please add reference for this.
2. line 219-2221 " whereas the role of MoDCs in immunity 219 induction is not fully understood. Because MoDCs are inflammatory DCs and involved 220 in the pathogenesis of autoimmune diseases, the immune response of MoDCs to immun-221 ization may lead to unfavorable side effects in neonates. Can authors supply reference for this role of moDC are inflammatory DCs ?
3. Line 116-120, " In clinical settings,... infants in developed countries." This is very strong statement and should be tone down and the authors should add references for these studies.
4. Line 133-138, Therefore, new technology that enhances the immune responses against vaccine antigens in neonates and infants is needed"... The authors need to discuss briefly what are the down sides of commercially available adjuvants such as MF59 or etc which have good safety profiles in human and immunocompromised individuals.

Author Response

Reviewer 2

<Comment 1>  Line 48-49 does not reflect the features of the immune system of human children. Please add reference for this. 

<Reply to the comment> According to the comment, we added the related references [22-24].

  1. Gruber W.C.; Darden P.M.; Still J.G.; Lohr J.; Reed G.; Wright P.F. Evaluation of bivalent live attenuated influenza A vaccines in children 2 months to 3 years of age: safety, immunogenicity and dose-response. Vaccine 1997, 15, 1379-84, doi: 10.1016/s0264-410x(97)00032-7.
  2. Gans H.A.; Arvin A.M.; Galinus J.; Logan L.; DeHovitz R.; Maldonado Y. Deficiency of the humoral immune response to measles vaccine in infants immunized at age 6 months. JAMA 1998, 280, 527-32, doi: 10.1001/jama.280.6.527.
  3. Wright P.F.; Karron R.A.; Belshe R.B.; Thompson J.; Crowe J.E. Jr; Boyce T.G.; Halburnt L.L.; Reed G.W.; Whitehead S.S.; Anderson E.L.; Wittek A.E.; Casey R.; Eichelberger M.; Thumar B.; Randolph V.B.; Udem S.A.; Chanock R.M.; Murphy B.R. Evaluation of a live, cold-passaged, temperature-sensitive, respiratory syncytial virus vaccine candidate in infancy. J Infect Dis 2000, 182, 1331-42, doi: 10.1086/315859.

<Comment 2>  line 219-221 " whereas the role of MoDCs in immunity induction is not fully understood. Because MoDCs are inflammatory DCs and involved in the pathogenesis of autoimmune diseases, the immune response of MoDCs to immunization may lead to unfavorable side effects in neonates. Can authors supply reference for this role of moDC are inflammatory DCs ? 

<Reply to the comment> According to the comment, we added the related references [103-106].

  1. Domínguez P.M.; Ardavín C. Differentiation and function of mouse monocyte-derived dendritic cells in steady state and inflammation. Immunol Rev 2010, 234, 90-104. doi: 10.1111/j.0105-2896.2009.00876.x.
  2. Terry R.L.; Getts D.R.; Deffrasnes C.; van Vreden C.; Campbell I.L.; King N.J. Inflammatory monocytes and the pathogenesis of viral encephalitis. J Neuroinflammation 2012, 9, 270. doi: 10.1186/1742-2094-9-270.
  3. Ge Z.; Da Y.; Xue Z.; Zhang K.; Zhuang H.; Peng M.; Li Y.; Li W.; Simard A.; Hao J.; Yao Z.; Zhang R. Vorinostat, a histone deacetylase inhibitor, suppresses dendritic cell function and ameliorates experimental autoimmune encephalomyelitis. Exp Neurol 2013, 241, 56-66. doi: 10.1016/j.expneurol.2012.12.006.
  4. Ko H.J.; Brady J.L.; Ryg-Cornejo V.; Hansen D.S.; Vremec D.; Shortman K.; Zhan Y.; Lew A.M. GM-CSF-responsive monocyte-derived dendritic cells are pivotal in Th17 pathogenesis. J Immunol 2014, 192, 2202-9. doi: 10.4049/jimmunol.1302040.

<Comment 3> Line 116-120, " In clinical settings,... infants in developed countries." This is very strong statement and should be tone down and the authors should add references for these studies. 

<Reply to the comment> We deeply appreciate the reviewer's constructive comments. We have tone down the statement focusing on the nutritional deficiency and newly added the possible influence of genetic and underlying disease burdens on the reduced effects of rotavirus vaccine by citing several literatures as following (lines 203-213):

In clinical settings, both the Rotateq and Rotarix vaccines have been shown to be safe, immunogenic, and highly protective against severe rotavirus gastroenteritis in infants in developed countries (e.g., North Americaand Europe); however, their effectiveness was much lower in Africa and Asia [50, 51]. Because vitamin A deficiency impairs the induction of acquired immune responses to those live-attenuated rotavirus vaccines in animal model [52], nutritional insufficiency (e.g., vitamin A) may be one of the hypothesized mechanisms on the reduced vaccine efficacy in infants in developed countries. On the other hand, a recent study demonstrated a genotype-dependent manner in the susceptibility to rotavirus infection and has reported the Lewis A phenotype is a restriction factor for Rotateq and Rotarix vaccine responses in Nicaraguan children [53]. In addition, another study elucidated that environmental enteropathy and malnutrition were associated with oral vaccine (Rotarix and Polio) failure in infants in Bangladesh [54]. Taken together, various factors (e.g., vitamin A deficiency, genotype and the burden of environmental enteropathy) need to be taken into consideration for the use of vaccine in infants in developed countries.

<Added new references>

  1. Chattha K.S.; Kandasamy S.; Vlasova A.N.; Saif L.J. Vitamin A deficiency impairs adaptive B and T cell responses to a prototype monovalent attenuated human rotavirus vaccine and virulent human rotavirus challenge in a gnotobiotic piglet model. PLoS One 2013, 8, e82966. doi: 10.1371/journal.pone.0082966.
  2. Bucardo F.; Nordgren J.; Reyes Y.; Gonzalez F.; Sharma S.; Svensson L. The Lewis A phenotype is a restriction factor for Rotateq and Rotarix vaccine-take in Nicaraguan children. Sci Rep 2018, 8, 1502. doi: 10.1038/s41598-018-19718-y.
  3. Naylor C.; Lu M.; Haque R.; Mondal D.; Buonomo E.; Nayak U.; Mychaleckyj J.C.; Kirkpatrick B.; Colgate R.; Carmolli M.; Dickson D.; van der Klis F.; Weldon W.; Steven Oberste M.; PROVIDE study teams.; Ma J.Z.; Petri W.A. Jr. Environmental Enteropathy, Oral Vaccine Failure and Growth Faltering in Infants in Bangladesh. EBioMedicine 2015, 2, 1759-66. doi: 10.1016/j.ebiom.2015.09.036.

<Comment 4> Line 133-138, Therefore, new technology that enhances the immune responses against vaccine antigens in neonates and infants is needed"... The authors need to discuss briefly what are the down sides of commercially available adjuvants such as MF59 or etc which have good safety profiles in human and immunocompromised individuals.

<Reply to the comment> We deeply appreciate the reviewer's constructive comments. According to the comments, we added the sentences regarding the recent and classical adjuvants (MF59 and alum) as following(lines 230-242):

    To date, various adjuvants (e.g., alum, chitosan and MF59) have been developed and used in mice and humans [62-67], and some of them were tested in children [64,66,67]. Alum, salts of aluminum, is the classical adjuvant most often used in vaccines in humans. A recent study used alum as an adjuvant for enterovirus 71 vaccine in healthy children aged 6-35 months and reported the safety and effectiveness of the alum-adjuvant vaccine [64]. On the other hand, another study demonstrated in vitro that an adjuvant effect of alum was weak in neonatal DCs compared with those of BCG and TLR8 agonist [68]. MF59, a potent oil-in-water emulsion, has been developed and used as adjuvant for influenza vaccine and confirmed its safety and effectiveness in adult [65]. Recent studies also demonstrated that MF59 is effective and safe in young children 6-35 months as influenza vaccine adjuvant [66,67], whereas it needs further confirmation whether MF59 is safe and effective in children as a potent universal adjuvant for other vaccine antigens.

<Added new references>

  1. Xia Y.; Fan Q.; Hao D.; Wu J.; Ma G.; Su Z. Chitosan-based mucosal adjuvants: Sunrise on the ocean. Vaccine2015, 33, 5997-6010. doi: 10.1016/j.vaccine.2015.07.101.
  2. Neimert-Andersson T.; Binnmyr J.; Enoksson M.; Langebäck J.; Zettergren L.; Hällgren A.C.; Franzén H.;Lind Enoksson S.; Lafolie P.; Lindberg A.; Al-Tawil N.; Andersson M.; Singer P.; Grönlund H.; Gafvelin G. Evaluation of safety and efficacy as an adjuvant for the chitosan-based vaccine delivery vehicle ViscoGel in a single-blind randomised Phase I/IIa clinical trial. Vaccine 2014, 32, 5967-74. doi: 10.1016/j.vaccine.2014.08.057.
  3. Zhu F.C.; Meng F.Y.; Li J.X.; Li X.L.; Mao Q.Y.; Tao H.; Zhang Y.T.; Yao X.; Chu K.; Chen Q.H.; Hu Y.M.; Wu X.; Liu P.; Zhu L.Y.; Gao F.; Jin H.; Chen Y.J.; Dong Y.Y.; Liang Y.C.; Shi N.M.; Ge H.M.; Liu L.; Chen S.G.; Ai X.; Zhang Z.Y.; Ji Y.G.; Luo F.J.; Chen X.Q.; Zhang Y.; Zhu L.W.; Liang Z.L.; Shen X.L. Efficacy, safety, and immunology of an inactivated alum-adjuvant enterovirus 71 vaccine in children in China: a multicentre, randomised, double-blind, placebo-controlled, phase 3 trial. Lancet 2013, 381, 2024-32. doi: 10.1016/S0140-6736(13)61049-1.
  4. O'Hagan D.T.; Rappuoli R.; De Gregorio E.; Tsai T.; Del Giudice G. MF59 adjuvant: the best insurance against influenza strain diversity. Expert Rev Vaccines 2011, 10, 447-62. doi: 10.1586/erv.11.23.
  5. Patel M.M.; Davis W.; Beacham L.; Spencer S.; Campbell A.P.; Lafond K.; Rolfes M.; Levine M.Z.; Azziz-Baumgartner E.; Thompson M.G.; Fry A.M. Priming with MF59 adjuvanted versus nonadjuvanted seasonal influenza vaccines in children - A systematic review and a meta-analysis. Vaccine 2020, 38, 608-619. doi: 10.1016/j.vaccine.2019.10.053.
  6. Esposito S.; Fling J.; Chokephaibulkit K.; de Bruijn M.; Oberye J.; Zhang B.; Vossen J.; Heijnen E.; Smolenov I. Immunogenicity and Safety of an MF59-adjuvanted Quadrivalent Seasonal Influenza Vaccine in Young Children at High Risk of Influenza-associated Complications: A Phase III, Randomized, Observer-blind, Multicenter Clinical Trial. Pediatr Infect Dis J 2020, 39, e185-e191. doi: 10.1097/INF.0000000000002727.
  7. Dowling D.J.; Scott E.A.; Scheid A.; Bergelson I.; Joshi S.; Pietrasanta C.; Brightman S.; Sanchez-Schmitz G.;Van Haren S.D.; Ninković J.; Kats D.; Guiducci C.; de Titta A.; Bonner D.K.; Hirosue S.; Swartz M.A.;Hubbell J.A.; Levy O. Toll-like receptor 8 agonist nanoparticles mimic immunomodulating effects of the live BCG vaccine and enhance neonatal innate and adaptive immune responses. J Allergy Clin Immunol2017, 140, 1339-1350. doi: 10.1016/j.jaci.2016.12.985.